# Detection of inundation areas due to the 2015 Kanto and Tohoku torrential rain in Japan based on multi-temporal ALOS-2 imagery

Wen Liu[1], Fumio Yamazaki[1]

[1] Department of Urban Environment Systems, Chiba University, Chiba, 263-8522, Japan

*Correspondence to*: Wen Liu (wen.liu@chiba-u.jp)

**Abstract.** Torrential rain triggered by two typhoons hit the Kanto and Tohoku regions of Japan from September 9 to 11, 2015. Due to the record-breaking amount of rainfall, several river banks were overflowed and destroyed, causing floods over wide areas. The PALSAR-2 sensor onboard the ALOS-2 satellite engaged in emergency observations of the affected areas during and after the heavy rain. Two pre-event and three co-event PALSAR-2 images were employed in this study to extract flooded

areas in Joso city, Ibaraki prefecture. The backscattering coefficient of the river water was investigated first using the PALSAR-2 intensity images and a land-cover map with a 10-m resolution. The inundation areas were then extracted by setting threshold values for backscattering from water surfaces in the three temporal Synthetic Aperture Radar (SAR) images. The extracted results were modified by considering the land-cover and a digital elevation model (DEM). Next, the inundated built-up urban areas were extracted from the changes in SAR backscattering. The results were finally compared with those from

visual inspections of airborne imagery by the Geospatial Information Authority of Japan (GSI), and more than 85% of the maximum inundation areas were extracted successfully.

## 1 Introduction

Floods are natural hazards that occur in most countries in the world. According to the special report on "Managing the risks of extreme events and disasters" from the Intergovernmental Panel on Climate Change (IPCC), increasing flood risks are a

20 concern due to global warming (Field et al., 2012). Floods can be classified as river (fluvial) floods, flash floods, urban floods, pluvial floods, sewer floods, coastal floods, and glacial lake outburst floods. Floods can also be categorized by their durations: flash floods, rapid onset floods, and slow onset floods (Dolan, 1995). Flash floods are the most lethal and are usually caused by heavy rainfall, tropical storms or dam failures. The water rushes quickly over land, flooding houses and destroying roads (Jonkman, 2005; Haynes et al., 2009). Rapid onset floods last for relatively longer periods of one or two days. Slow onset

floods may last for weeks or even months. This type of flood occurs almost every year in Thailand. In 2011, the worst flooding of the last five decades hit Thailand from August 5 to January 9, lasting 158 days (Gale and Saunder, 2013; Nakmuenwai et al., 2017).

Remote sensing using satellite imagery is effective for repeatedly observing broad areas on the Earth's surface. There are two categories of remote sensing based on sensor type: passive (mainly optical and thermal) and active (mainly radar). Optical

sensors work only in daytime and cannot observe objects under cloud-cover conditions. Radar sensors such as Synthetic Aperture Radars (SARs) can avoid this problem. SAR systems have been used widely in disaster situations such as earthquakes, volcanic eruptions, tsunamis, typhoons and floods (Weissel et al., 2004; Feng et al., 2013; Liu et al., 2013; Dumitru et al., 2015; Yulianto et al., 2015). Klemas (2015) and Lin et al. (2016) summarized recent research on flood assessments using

optical and SAR sensors. Because microwaves exhibit specular reflections against a smooth water surface, water regions in a SAR image show low backscattering intensity. SAR images are effective for extracting inundation areas. Several methods, both pixel- and object-based, have been proposed to extract inundation zones from SAR images (Martinis et al., 2009 and 2013; Hoque et al., 2011; Manjusree et al., 2012; Pulvirenti et al., 2014; Kundu et al., 2015; Nakmuenwai et al., 2017). Thresholding is a common and effective pixel-based approach. Since backscattering of a water surface depends on many

factors such as acquisition conditions of SAR images and their environments, its value is highly variable. It is difficult to judge the most suitable value objectively without additional information. Automated thresholding methods using the gray-level histogram have been introduced to overcome this issue (Fan and Lei, 2009; Martinis 2009 and 2013; Pulvirenti et al., 2011; Nakmuenwai et al., 2017). The global threshold value was merged from several local threshold values, which were obtained from the multimodal histograms of sub-areas. However, this approach is time-consuming when the study area is large. In

addition, sufficient contrast was necessary for automated thresholding. Giustarini et al. (2013) found the previous proposed methods were difficult to work in urban areas containing radar shadow and layover. They proposed a method based on image differencing to detect floodwater inside urban areas. Mason et al. (2009 and 2012) used a SAR simulator and Lidar data to estimated inundated buildings. Interfeormatic coherence was also used to extract floods in either rural or urban areas, but the acquisitions of temporal and spatial baselines were strict (Nico et al. 2000; Chini et al. 2012; Pulvirenti et al. 2016). All of

these researches used SAR images taken by X- and C- bands with short wavelengths, which were sensitive to separate water and non-water regions. Flood mapping using L-band satellite images was few (Zhang and Wang 2003; Allan et al., 2012; Yulianto et al. 2015).

Limited by long revisit cycles, satellite images have been used more for post-flood analysis than for monitoring floods (Jain et al., 2005). ALOS-2 was launched on May 24, 2014 and is a follow-up satellite of the ALOS program. It carries the PALSAR-

2 enhanced high-resolution SAR sensor. Owing to the right-and-left looking function of the PALSAR-2 sensor, the observation repetition frequency is improved. It is now possible to monitor affected areas shortly after a disaster strikes (JAXA, 2017a). PALSAR-2 images have been used to detect damage following the 2015 Gorkha, Nepal, earthquake (Watanabe et al., 2016) and to detect pyroclastic ash coverage on Kuchinoerabu Island, Japan (Hara et al., 2016; Natsuaki et al., 2017). The 2015 Kanto and Tohoku torrential rain is the first flood event over a wide range in Japan after the ALOS-2 launched. PALSAR-2

performed emergency observations of the impacted areas during and after the heavy rain (Natsuaki et al. 2016; Kwak et al., 2017; Rimba and Miura 2017).

In this paper, five pre- and co-event PALSAR-2 images are employed to monitor the changes in the inundation areas in Joso city, Ibaraki prefecture, Japan. The images were used in a previous study to extract the inundations (Yamazaki and Liu, 2016). In the study, one threshold value of backscattering intensity was investigated using the pre-event water regions and the pre-

event PALSAR-2 images, and it was applied to all co-event images. In addition, the obtained results were only verified via visual comparison. In this study, the method of the inundation extraction is improved by introducing land-cover information and elevation data. The flooded urban areas are also extracted using the intensity difference between the pre- and co-event images. The obtained results are verified quantitatively via comparison with those from visual inspections of airborne imagery.

**2 The study area and dataset**

Affected by two typhoons, torrential rain hit the Kanto and Tohoku regions of Japan from September 9 to 11, 2015, and destructive floods were caused in many places. A linear heavy-rain cloud was generated on September 9 and moved slowly from Kanto to Tohoku. It remained in the upstream region of the Kinugawa river for several hours, as shown in **Figure 1(a)** (CEReS, 2015). The maximum cumulative rainfall exceeded 600 mm in the Kanto region and 500 mm in the Tohoku region,

which are record-breaking volumes in those parts of Japan. Due to the rising water levels, collapsed banks and overflows were reported for 85 rivers (Cabinet Office, Government of Japan, 2016).

Joso city is located approximately 50 km to the north-west of Tokyo, as shown in **Figure 1(b)**. In the figure, the study area is depicted by the red rectangle, where the Kinugawa and Kokai rivers flow from north to south. The locations of the two rivers are shown in **Figure 2**. Due to the heavy rainfall, the water volume of the Kinugawa river increased rapidly in Joso city in the

15 early morning of September 10, 2015. An overflow of the river bank in the Wakamiyado district (yellow square I in **Figure 3**) was reported by the city government at 7:40 am. The flood water flowed through the city from north to south. A river bank failure finally occurred in the Misaka district (yellow square II in Figure 3) at 12:50 in the afternoon, and flood waters quickly covered almost the entire area between the two rivers.

**Table 1** shows the observational conditions for the five PALSAR-2 images used in this study. The radar incidence angles are

20 almost the same, 39.7°, at the centers of the images. However, the paths are grouped into two ascending and one descending parts; paths A and C have a 344° heading angle clockwise from the north, and path B has a 195° heading angle, as shown in **Figure 1(b)**. The ascending paths observed the target area at nighttime, but the descending path occurred in daytime at just before noon in Japan. Paths B and C are right-looking, whereas path A is left-looking. The images were all acquired with HH polarization and in the Ultra-fine mode (JAXA, 2017a). The images for Path A were the same data used in the study of Rimba

and Miura (2017). The five datasets were provided as ranges and single-look azimuths compressed at a processing level of 1.1, which is represented by the complex I and Q channels to preserve the amplitude and phase information (JAXA, 2017a).

ENVI/SARscape software was used, and several pre-processing steps were applied. A multi-look process with two looks was applied in the range and azimuth directions to improve the quality of the SAR images and maintain the resolution as much as possible; the subsequent azimuth resolution was 4.2 m, and the slant range resolution was 2.9 m. A 5-m digital elevation model

(GSI, 2017a) was employed to project the data onto a WGS84 reference ellipsoid with a pixel size of 2.5 m. Radiometric calibration was carried out to convert the amplitude data into backscattering coefficient (sigma naught) values (JAXA, 2017a). An enhanced Lee filter with 5×5 pixels was applied to reduce the speckle-noise while keeping the details. Because the images

taken in path A did not cover the entire target area, a mask was applied to the other images to leave only the common area. The pre-processed backscattering coefficient images are shown in **Figure 3**. Compared to the pre-event images, a decrease in backscattering intensity can be confirmed between the Kinugawa and Kokai rivers from the images taken on September 11 and 13. The decrease was caused by specular reflection from the water surface, which also indicates the inundation. The image taken on September 11, after the bank broke, shows the lowest backscatter in the target area.

The 5-m digital elevation model (DEM) is shown in **Figure 2(a)**. It was created from Lidar data with standard deviations of less than 1.0 m in the vertical direction and 0.3 m in geolocation (GSI, 2017a). Most of the target area is flat, with an elevation difference of less than 20 m, especially for Joso city, which is located between the rivers. The altitude gradually decreases from upstream (north) to downstream (south). A land-cover map was introduced to understand the surface conditions in the inundated area. The land-cover maps were produced by Hashimito et al. (2014), who used multi-temporal optical satellite data and were published by JAXA (2017b). The land surface was classified into 10 classes: water, urban, rice paddy, crop, grass, deciduous broad-leaved tree, deciduous needle-leaved tree, evergreen broad-leaved tree, evergreen needle-leaved tree and bare land. The land-cover map (version 16.09) for the target area is shown in **Figure 2(b)**. The classes of deciduous broad-leaved and needle-leaved trees were merged into one deciduous tree class, and a similar merge was applied to the evergreen tree classes. In addition to the Kinugawa and Kokai rivers, Sanuma lake, which is located in the north, is classified as permanent water. Most of the study area is covered by rice-paddies and crop fields. Between the rivers, three large settlement areas exist, and they were classified as urban. The inundation map produced by the Geospatial Information Authority of Japan (GSI) (GSI, 2015) is shown in **Figure 2(c)**. It was made by visual interpretations of multi-temporal aerial photographs. The 40-km2 area within the blue polyline was estimated as the inundation area on September 10, after the bank collapsed. The change in inundation area from September 11 to 16 is shown by the filled polygons. This inundation map was used as the truth data to verify our extraction results.

**3 The field survey in Joso city**

A field survey was carried out on October 26, 2015, one month after the heavy rainfall. The route of our survey is shown in **Figure 4(a)**, overlapped on Google Earth. Recovery work was ongoing, and several roads were still closed. The overflow location (I) in the Wakamiyado district and the collapsed bank location (II) in the Misaka district were primarily investigated. The pre- and co-event aerial photographs, which were taken by GSI (GSI, 2017b), are shown in **Figures 4(b, c)**. The pre-event images were taken in 2007, and the co-event images were taken on September 11, 2015, after the overflow and bank collapse occurred. There were two groups of solar panels in the Wakamiyado district, which were located next to the Kinugawa river. A part of the natural levee in this area, as shown in the pre-event aerial photo, was excavated in March 2015 to set up solar panels. A large amount of water flowed from the Kinugawa river and washed away the solar panels. The residential areas and farmlands in Joso city were widely flooded, which can be observed clearly in the co-event aerial photo. During the field survey, the removal of the flooded solar panels was underway. Many panels were still scattered on the farmland, as shown in **Figure**

5(a). The temporary bank, which was built up using concrete blocks and sand bags for emergency restoration, can be seen in the first photo. The width of the collapsed bank in the Misaka district was 20 m at first and gradually expanded to a final width of 200 m. Many wooden houses behind the bank were washed away. Approximately 1/3 of the area of Joso City was inundated by the flood. As part of the emergency recovery work, temporary foot protection blocks were installed on September 16. Additional reinforcement work using a steel sheet-pile was performed outside the provisional bank after the main reinforcement work was finished on September 19. The double wall cofferdam with the sheet-pile (left) and the filling bank (right) can be identified in the left photo in **Figure 5(b)**. Due to the rapid flow from the Kinugawa river, the telegraph poles tilted, and the road was blocked. The damaged houses could still be seen in October during our field survey.

The color composites of the pre-processed SAR intensity images for these two locations are shown in **Figure 4(d)**. The top image shows the Wakamiyado district; the pre-event image taken on August 13 is shown in green and blue, and the co-event image on September 10 is shown in red. The cyan pixels represent the decrease in backscatter, which also indicates the flooded area. As the water level rose in the Kinugawa river, the island in the river and the dry riverbed were under water. The overflow location can be confirmed on the right side of the river. The bottom image shows the Misaka district; the pre-event image taken on July 31 is shown in green and blue, and the co-event image on September 11 is in red. Due to the bank collapse, the houses were washed away and show a decrease in backscatter. Thus, the location can be confirmed easily from the cyan color.

## 4 Thresholding method for water region

In this study, the threshold value for water was investigated automatically using reference areas. The white and black polygons shown in the enlargement of land-cover of **Figure 2(b)** were used as water and non-water references, respectively. The references were selected according to the aerial photos and the land-cover map. Water references over a total area of 0.24 km$^2$ were selected from Sanuma lake. Considering the changes in water levels in rivers, the Kinugawa and Kokai rivers were not used as water references. The non-water references were selected around the water references, outside of the inundation area. They include five different land-cover classes: urban, crop, deciduous tree, evergreen tree and bare land. The non-water references covered an area of 0.26 km$^2$. The histograms of backscattering intensity for the water and non-water references in the five SAR images are shown in **Figure 6** with respect to three path groups. The backscattering intensity of the non-water references was obviously higher than that of the water references. The mean values and the standard deviations (STD) for these references are summarized in **Table 2**. The images taken on the same path show similar backscattering characteristics for both the water and non-water references. However, the references show different backscatter characteristics for the different paths. Thus, the threshold value for water extraction should be determined for each image.

According to **Figure 6**, the threshold value of water was set from -15 dB to -10 dB, with 0.1-dB intervals. The extracted results were verified using the references. The optimal threshold value for each image with the maximum overall accuracy (O.A.) and Kappa coefficient was obtained and is shown in **Table 2**. Using the optimal threshold value, the O.A. for each of the five SAR images exceeded 94%, and the Kappa coefficients exceeded 0.89, which indicates that the water and non-water regions could

be distinguished correctly. The optimal threshold value should be obtained for each image; therefore, this approach is time-consuming to apply for many multi-temporal images.

Statistical features were introduced to calculate the threshold value in a simple manner. Because the backscattering intensity of the water references is more stable than that of the non-water regions, the mean and STD values of the water references were used to investigate the threshold value. Furthermore, the water references were commonly available using the GIS database. The results of combinations using the mean and STD values (mean+2×STD) of the water references are close to the optimal values (**Table 2**). Although the obtained values differ from the optimal value, only a limited decrease was seen in both O.A. and the Kappa coefficient.

## 5 Extraction of inundation areas

The extraction of inundation areas was conducted in three steps. First, the water regions were extracted from each SAR image. The land-cover map and the extracted results from the pre-event images were used to improve the results from the co-event images. The inundated urban areas were then detected by the difference in the backscattering coefficient (sigma naught) values between the pre- and co-event SAR images. Finally, the 5-m DEM was applied to modify the extracted inundation area. A flowchart of the current approach is shown in **Figure 7**.

### 5.1 Water extraction

First, the water regions in the two pre-event PALSAR-2 images were extracted using the threshold values proposed in the previous section. Extracted pixel groups that were smaller than 0.01 km$^2$ (1600 pixels) were removed as noise. The results obtained were overlapped on the land-cover map, as shown in Figure 7. The white pixels show non-water regions, and the colored pixels show regions covered by water. In addition to the Kinugawa and Kokai rivers, large paddy fields were extracted as water regions, especially from the image from August 13, 2015. This might have been caused by pouring water for rice cultivation. It is also possible that paddy fields hold rainfall more than other land-covers, but not much rainfall was recorded during the period of August 1-13, 2015 (Japan Meteorological Agency, 2017). Some parts of crop, grass and bare land land-covers were also extracted as water because the wavelength of L-band microwaves is 26 cm; these land-covers have little surface roughness and behave as specular reflectors, similar to water. Land-cover masks were created from the extracted smooth areas, excepting the water and rice paddy classes. The mask created from the image taken on July 31, 2015, represents the commission error for path A and that created from the image on August 13, 2015, represents the error for path B.

The water regions during and after the heavy rainfall were also extracted from the three SAR images taken in September 2015 using the proposed threshold values. After removing noise from the results, the land-cover masks were applied. For the September 10 image, the path B mask was applied to remove the extracted crop, grass and bare land areas. The path A mask was applied to the September 11 and 13 images. Although the image from September 13 was taken from a different path, the backscattering characteristics were similar to the image taken from path A owing to the same heading angle but opposite range

direction. The final results are shown in **Figure 8** (the three images on the right), for which most of the water commission errors due to smooth land-cover were reduced successfully. The extracted water regions in the co-event images were mainly water and rice paddy class land-covers.

Due to the rise in water level on September 10, the width of the Kinugawa river doubled from that of July and August. The high-water riverbeds covered by grasses and crops were extracted as water due to inundation. The total area of 27.1 km$^2$ was extracted as water, but 6% of it was crop and 2% was vegetation. In the SAR images from September 11 and 13, the water level and river width returned to their original situations. By observing the extracted water regions between the main rivers, the inundation, which was primarily in the north on September 10, was caused by the overflow in the Wakamiyado district. It expanded to the south on September 11 after the bank collapsed in the Misaka district. A total area of 24.7 km$^2$ was extracted as water on September 11, including 5% crop and 1% grass areas. On September 13, the water receded in the north area but still occupied the south area, which agreed with the visual inspection by GSI (2015). A total area of 16.3 km$^2$ was extracted as water. Because the land-cover mask applied to this image was made for path A, which does not completely match the SAR data, the commission errors due to the smooth land-covers (crop and grass) accounted for 15% of the extracted regions.

### 5.2 Inundated urban areas

Different from other land covers, the backscattering intensity of urban areas still showed high values after inundation owing to multiple reflections from buildings and water surfaces (Mason, 2010 and 2012; Kwak et al., 2017). Thus, it is difficult to extract the inundated urban areas using the proposed simple water threshold values. In this study, backscatter differences were added to extract inundated urban areas. The differences in backscattering intensity for paths A and B were obtained and are shown in **Figure 9(a)**. The mean and STD values for the urban areas within the non-water references were obtained. The mean value of the difference in backscatter for path A was 0.10 dB, whereas the STD value was 1.28 dB. For path B, the mean value was 0.82 dB, and the STD value was 1.72 dB. The threshold value for the inundated urban areas was set using the same method as for water, i.e., by using the mean and STD values (mean + 2 STD). Therefore, the urban area where the backscatter increased more than 4.28 dB for path A was extracted as being under inundation on September 10, and that exceeding 2.66 dB for path B was extracted as being under inundation on September 11.

### 5.2 Verification and improvement

The extracted water regions, except the Kiugawa and Kokai rivers on September 10, 11 and 13, and the extracted urban areas with increased backscatter on September 10 and 11 were considered to be the area of inundation. To verify this result, the inundation map shown in **Figure 2(c)** was introduced as the truth data. The truth data focused on the plain in Joso city between the rivers; the extracted results within the black dotted frame of **Figure 9(a)** were enlarged and are shown in **Figure 8(b)** for comparison with the truth data.

In **Figure 9(b)**, the extracted inundation area is primarily paddy fields and urban land cover. According to the GSI, a total area of 40 km$^2$ was estimated to have been affected by the heavy rainfall in this area, as the blue polyline shows for the result of

September 10. Because the SAR image on September 10 was taken before the bank collapse, the inundation in the southern part of the Wakamiyado district was not extracted. At 1:00 pm (JST) on September 11, the inundation area reached its maximum, 30.5 km$^2$, as the brown polyline shows for the result on the same day. Compared with the truth data, our result at 10:00 pm almost matched the polyline of GSI, which included an area measuring 21.7 km$^2$. Part of the residential area around
Joso City Hall was successfully extracted as inundation. However, several urban areas and roads that were under water could not be extracted. The truth data on September 13 were reported at 10:40 am (JST) for an area of 15.2 km$^2$ as the yellow polyline on the result of the same day. Because the SAR image of September 13 was taken from path C without a pre-event image, the inundated urban area could not be extracted in this study. However, the extracted inundated paddy fields were close to the truth data. The flooded water in the Wakamiyado district disappeared owing to the drainage work. However, the Misaka and
Mitsukaido districts (around City Hall) were still flooded. The extracted inundation area at 11:37 pm on September 13 was 10.7 km$^2$.

The comparison of the extracted inundation and the truth data within the estimated affected area is shown in **Table 3**. The producer accuracy for the result on September 11 was 65.6%, whereas the user accuracy was 92.2%. The low producer accuracy was caused by the inundated urban areas and roads that could not be detected by either low or increased backscatter.
The drainage work was also considered to be a reason for the low producer accuracy. The overall accuracy (O.A.) was 68.7%, and the kappa coefficient was 0.33. The producer accuracy for the result for September 13 was 61.0%, and the user accuracy was 87.0%, both of which were lower than those obtained for September 11. However, the O.A. was 81.3%, and the kappa coefficient was 0.58, which were both higher than the results for September 11.

In the previous study (Yamazaki and Liu, 2016), the inundation areas in the three co-event PALSAR images were extracted
using one threshold value of -12.4 dB, which was estimated by comparing the backscatter intensity for the original water regions (Kinugawa and Kogai rivers, Sanuma lake) and the other areas in the whole study area. As a result, 20.4 km$^2$ on September 11 and 16.3 km$^2$ on September 13 were extracted as inundation. Since the threshold values used in this study were -13.5 dB for September 11 and -14.2 dB for September 12, respectively, lower than that of the previous study, the extracted areas including the inundated built-up areas were similar in size to that of the previous results. However, the producer and user
accuracies increased 3%, whereas the O.A. increased 2% for the result on September 11. For the result on September 13, the producer accuracy decreased whereas the user accuracy increased from 68.8% to 87%. The O.A. increased significantly from 77.4% to 81.3%, while the kappa coefficient increased from 0.53 to 0.58. The individual threshold values for the images taken in different acquisition conditions were more effective than one common value.

The 5-m DEM was introduced to improve the inundation extraction result and estimate the inundation depth on September 11.
The elevation between the Kinugawa and Kokai rivers, which is shown as the black arrows in **Figure 3(a)** and **Figure 9(b)**, is shown in **Figure 10(a)**. The blue arrow represents the initial extracted result for September 11. It shows that the low altitude plain area between the rivers was extracted as inundation. The inundation height on the Kinugawa river side was 50.4 m, whereas that on the Kokai river side was 50.2 m. Considering the flow of water, the inundation heights on the east and west sides should be almost equal. Therefore, the flooded range was modified to match the higher inundation height. The modified

inundation is shown as the red arrow. The inundation depth was calculated by subtracting the altitude of the ground surface from the inundation height.

This modification was carried out from north to south, which is the rivers' transverse direction. To remove the influence of bad DEM values, a low pass filter with a $3 \times 3$ pixel window was applied to smooth the elevation data. The modified result for September 11 is shown in **Figure 10(b)**. The final estimated area of inundation was 27.3 km$^2$, which is closer to the value of 30.5 km$^2$ given in the truth data than to the initial result. The inundation heights are also shown in **Figure 10(b)** by the color bar. According to the obtained inundation heights and areas, the total inundation volume was estimated as 3.4×107 m$^3$. Most of the inundation depths were approximately 1 m; the maximum depth was found to the northwest of City Hall. The estimated inundation depths show good agreement with the truth data, where the deeper areas were inundated for a longer time. The most deeply flooded area was still inundated on September 16. After the modification, the producer accuracy increased to 79.6%. The O.A. and kappa coefficient were also increased to 77.0% and 0.43, respectively, which show the effects of this modification.

An enlargement of the area surrounding City Hall is shown in **Figure 11**. According to the aerial photo in **(b)**, this area was still flooded on September 11, 2015. The increase in the backscattering intensity in the SAR image could be confirmed from the color composite **(a)**, especially in the parking lot, which was caused by the multiple reflections of the vehicles and the water. Water marks were observed in the field survey, as shown in the ground photo **(c)** taken in front of City Hall. According to the water marks, the maximum inundation height was approximately 1.2 m, and the sustained water level was approximately 0.6 m, which shows good agreement with the estimated inundation height of 57 cm by our analysis.

## 6 Discussions

To verify the effectiveness of our results, a comparison with the previous studies for the same event was carried out. Natsuaki et al. (2016) proposed a combination of coherence and amplitude values to detect affected areas using two pre-event and one co-event PALSAR-2 images taken on September 12, 2015. Inundation was extracted by the decrease of coherence and a low backscatter intensity. Kwak et al. (2017) extracted the floods on September 11 from a pair of pre-event and co-event PALSAR-2 images taken on July 31 and September 11, 2015, which were also used in our study. Flooded rice paddies were extracted by the differences of intensity whereas flooded urban areas were extracted by the correlation coefficient. These two researches extracted both the inundated rice paddies and urban areas using only SAR images. The producer accuracy in the study of Natsuaki et al. (2016) was 75%, a little higher than our results before the improvement using DEM. However, the O. A. was 52% since some areas could not be evaluated due to low pre-event coherence values. Our method using only the backscatter intensity could be applied to the whole study area. The accuracy in the study of Kwak et al. (2017) was not indicated. By visual comparison, their results extracted more inundated areas with more commission errors. Many agriculture fields outside the inundation were extracted as false alarms.

Rimba and Miura (2017) compared three common methods, unsupervised and supervised classifications, threshold method using the same SAR pair of Kwak et al. (2017) and 5-m DEM. The scheme of threshold method showed the best result, which extracted water regions from the pre- and co-event images, respectively, similar with our proposed approach. The inundation was obtained by the change of the extracted water regions. However, their scheme would not work when the inundated rice paddies were poured in a pre-event image. Our method could overcome this problem by applying the land cover map. The inundated urban areas were not extracted in the Rimba and Miura's research.

These three previous studies for this event were all based on change detection, which needs more than one pre- and post-event SAR pair. The threshold values used in these studies were defined by training samples, the same as our proposed method. Several common automated thresholding algorithms were applied to the PALSAR-2 image on September 11 to compare with our results (Kapur et al., 1985; Ridler and Calvard, 1978; Kittler and Illingworth, 1986; Otsu, 1979). Most of the automated algorithms extracted the inundation excessively. **Figure 12** shows the comparison of two best results by the Otsu (1979) and Minimum error thresholding algorithms (Kittler and Illingworth, 1986) and our thresholding result. In this enlarged region, the overall accuracy for the three results were calculated. Our results using the threshold value -13.5 dB obtained the highest accuracy as 74%, whereas that for the Otsu method was 55% and for the Minimum error method was 72%. After merging the extracted urban areas, the overall accuracy increased to 76%.

Although both the pre- and co-event SAR images were used in this study, the extraction of water region was carried out for each image. When a pre-event image is not available, an inundation map can still be created by the thresholding method. Inundated urban areas can be extracted by the change detection, however, a pre-event image taken in the same path is necessary. A land-cover map was introduced to define the threshold values and to reduce commission errors in a smooth surface due to a longer L-band wavelength. In the thresholding approach, the land cover could be replaced by visual interpretation. Without a land cover map, commission errors in the inundation extraction would decrease the accuracy. The 5-m DEM was used for improving the inundation map and for estimating the inundation depth. In an emergency response phase, our method could still obtain a reasonable result with the overall accuracy higher than 70%. Thus, our method is still valid if a land-cover map and detailed DEM data are not available. However, the accuracy of the obtained inundation map would decrease.

## 7 Conclusions

In this study, the flood situation in Joso city, Ibaraki prefecture, Japan, caused by heavy rainfall in September 2015 was monitored using five pre- and co-event ALOS-2 PALSAR-2 intensity images. The threshold value for water extraction was discussed using the pre-event images and a 10-m land-cover map. The water regions on September 10, 11 and 13 after the heavy rainfall were extracted using the proposed threshold values. The land-cover map was applied to reduce the commission errors caused by smooth ground surfaces (crop, grass and bare land). In addition, the differences in backscattering intensities were introduced to extract inundated urban areas, which showed high backscatter even in the inundation period. The result for September 10 shows that the inundation in the Wakamiyado district due to the bank overflow could be extracted correctly.

The expansion of the inundation area after the bank collapse was observed by the results for September 11 and 13. When compared against the truth data produced by GSI, most of the inundated area on the plain between the Kinugawa and Kokai rivers was extracted successfully, with an overall accuracy exceeding 60%.

The extracted results were improved by adding the 5-m resolution digital elevation model (DEM). The inundation area extracted for September 11 was modified and expanded to 9.0 km$^2$. The inundated area with objects (e.g., buildings and trees) higher than the inundation surface could be extracted by this modification. The overall accuracy of the extraction was increased from 64.1% to 77.0%. Comparing the estimated inundation depths and the GSI inundation map, the period of inundation was found to be longer for the area with high inundation depths.

Based on our results, the proposed thresholding method using water references is capable of extracting water regions from single SAR images. However, inundated urban areas could not be detected using this method. Although this problem was overcome in this study by introducing backscatter differences, the method will be difficult to apply to events without pre-event images. The modification using the DEM showed promising results, but several inundated urban areas and roads could still not be extracted. These omission errors were caused by high backscatter from the parts higher than the inundation surface. In this study, the 10-m land-cover map and the 5-m high-resolution DEM were introduced to improve the inundation extraction results. Without that additional information, the extraction accuracy might be decreased. In the future, the changes in gradient in the along river direction will be considered to improve accuracy when extracting inundation areas and depths.

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

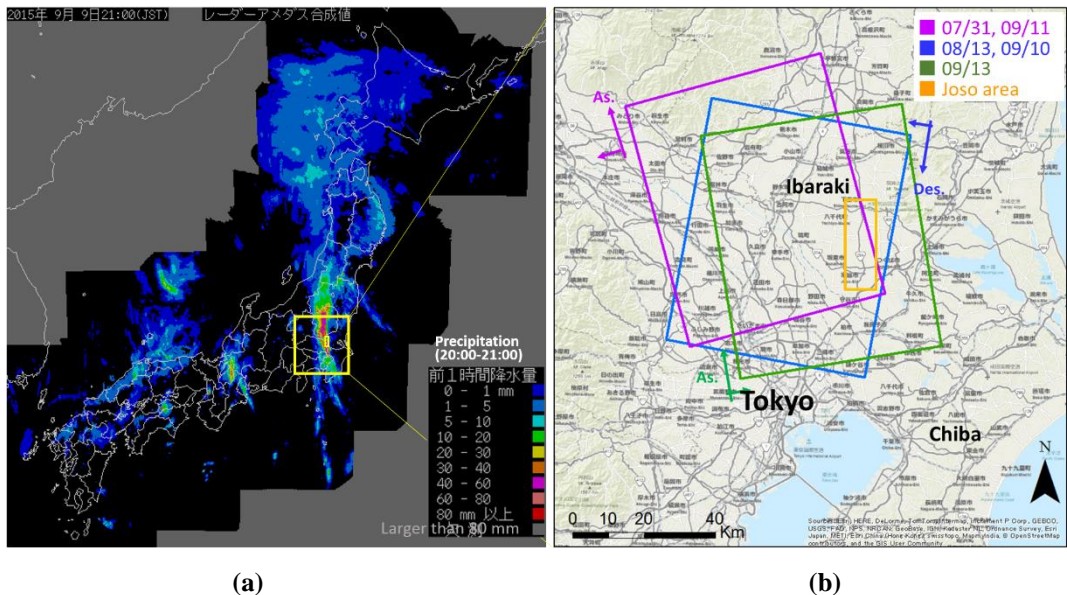

**(a)**  **(b)**

Figure 1: (a) One-hour rainfall distribution map at 21:00 on September 9, 2015, where a strip of heavy rainfall can be observed along the Kinugawa river catchment (CEReS, 2015); (b) coverage of five PALSAR-2 data sets in the Kanto region of Japan and the study area (Joso city), which is encompassed in the red square.

**Table 1: Acquisition conditions for the five PALSAR-2 images used in this study, which were taken from three different paths (A, B, and C).**

| Date | 7/31 | 8/13 | 9/10 | 9/11 | 9/13 |
|------|------|------|------|------|------|
| Time (JST) | 21 : 56 | 11:43 | 11:43 | 22:57 | 23:37 |
| Heading [°] | 344 | 195 | 195 | 344 | 344 |
| Look direction | left | right | right | left | right |
| Incident angle [°] | 39.7 | 39.7 | 39.7 | 39.7 | 39.7 |
| Path | A | B | B | A | C |

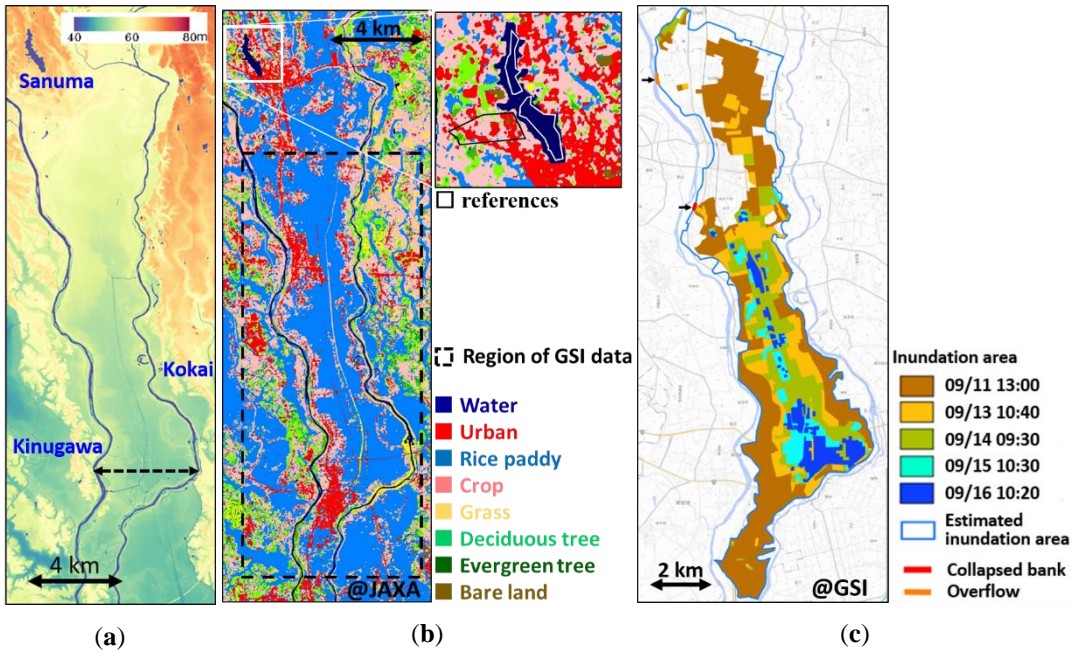

**Figure 2: (a) 5-m digital elevation model (DEM) produced by GSI (2017a); (b) 10-m land-cover classification produced by JAXA using ALOS AVINIR-2 images (JAXA, 2017b;Hashimoto et al., 2014); (c) reference data for the inundation areas produced by GSI by visual interpretations of aerial photographs (GSI, 2015), which cover the white dashed square in (b).**

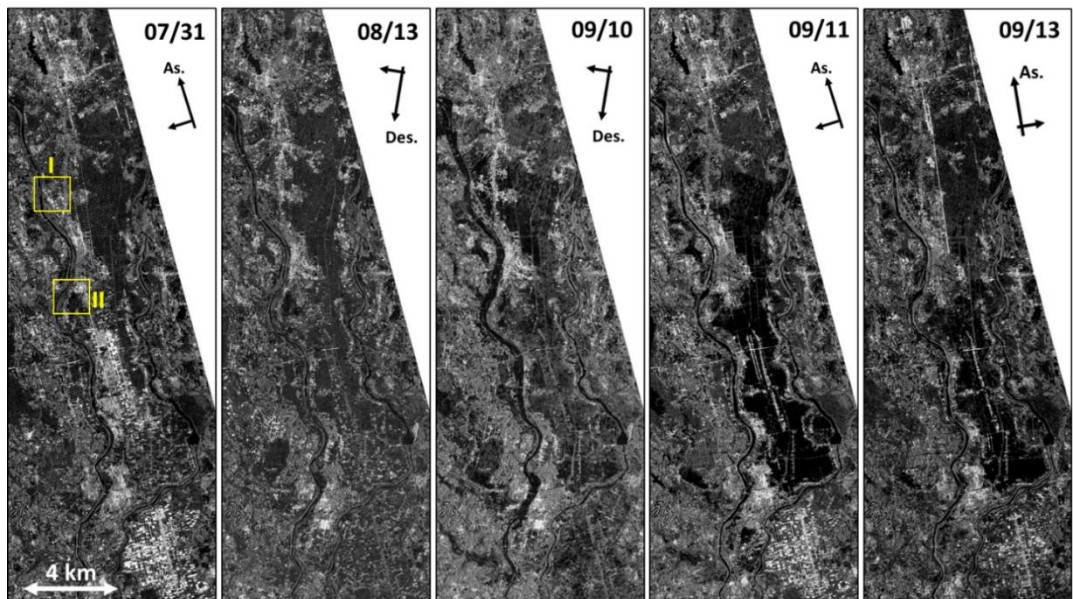

**Figure 3: Backscattering coefficient (sigma naught) images for the five temporal PALSAR-2 images after pre-processing.**

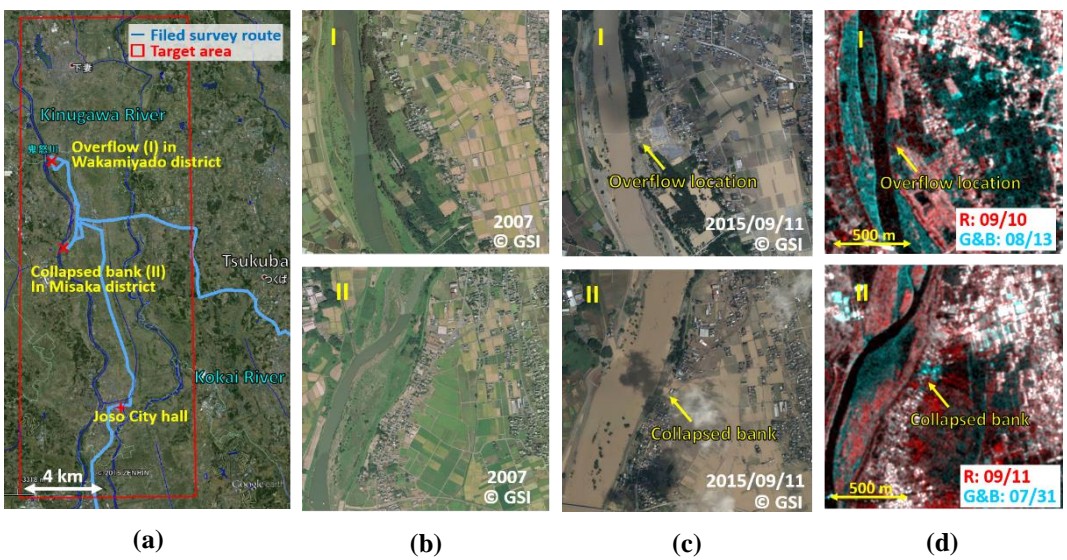

**Figure 4: (a) Route of the field survey (light-blue line) carried out by the authors on Oct. 26, 2015, and the locations of major damage; (b-c) aerial photographs taken in 2007 and on Sep. 11, 2015, by GSI at locations I and II (GSI, 2017b); (d) color composites of the multi-temporal PALSAR-2 images at locations I and II.**

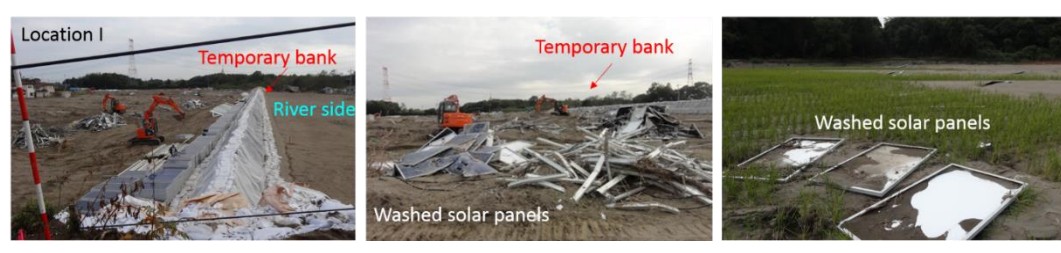

**(a) Wakamiyado district**

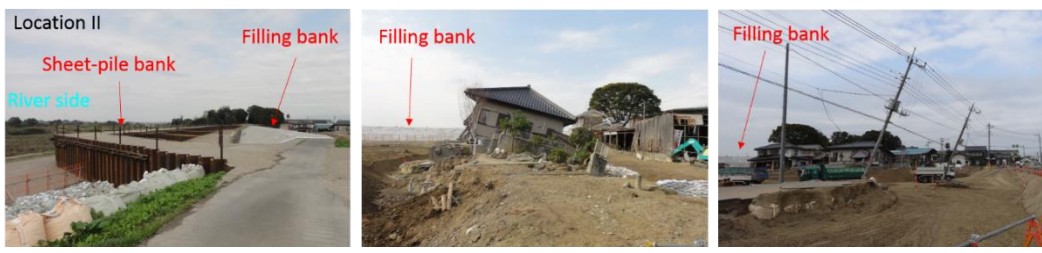

**(b) Misaki district**

Figure 5: Field photos taken on Oct. 26, 2015, in the Wakamiyado district at location I (a) and in the Misaka district at location II (b). Location I and II are shown in Figure 4.

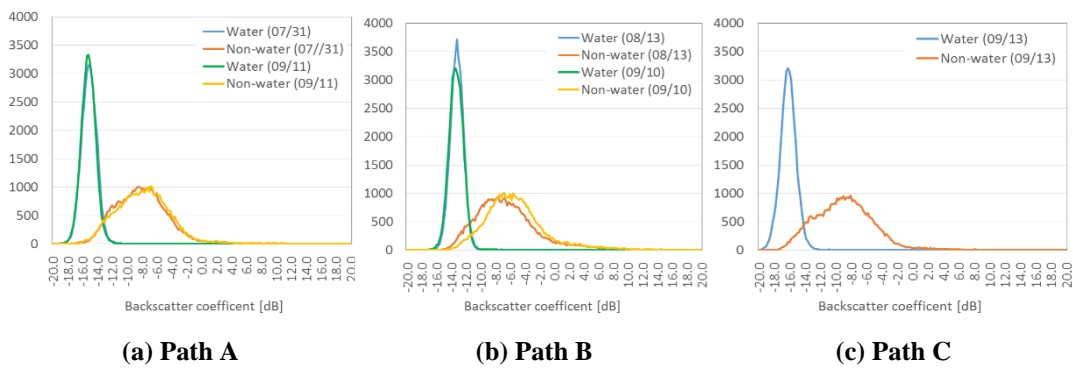

**(a) Path A**      **(b) Path B**      **(c) Path C**

**Figure 6: Comparison of the backscattering coefficient values within the water and non-water references from the PALSAR-2 images taken from different paths.**

**Table 2: Profiles of the backscattering coefficients for the water and non-water references; the threshold values were obtained by the optimal solution with best accuracy and by the proposed method (mean + 2 Std.)**

| Path | Date | Water: 39120 pixels | | Non-water: 41888 pixels | | Optimal solution | | | Mean + 2 STD | | |
|------|------|------|------|------|------|------|------|------|------|------|------|
| | | Mean | STD | Mean | STD | Threshold | O.A. | Kappa | Threshold | O.A. | Kappa |
| A | 7/31 | -15.20 | 0.99 | -8.20 | 3.47 | -13.2 | 95.9% | 0.92 | -13.2 | 95.9% | 0.92 |
| | 9/11 | -15.29 | 0.88 | -7.92 | 3.56 | -13.4 | 97.3% | 0.95 | -13.5 | 95.9% | 0.92 |
| B | 8/13 | -13.34 | 1.05 | -6.58 | 4.23 | -11.6 | 94.5% | 0.89 | -11.2 | 93.9% | 0.88 |
| | 9/10 | - 13.40 | 0.97 | -5.38 | 4.20 | -11.3 | 96.8% | 0.94 | -11.4 | 96.8% | 0.94 |
| C | 9/13 | -16.26 | 1.04 | -8.98 | 3.83 | -14.2 | 94.7% | 0.89 | -14.2 | 94.7% | 0.89 |

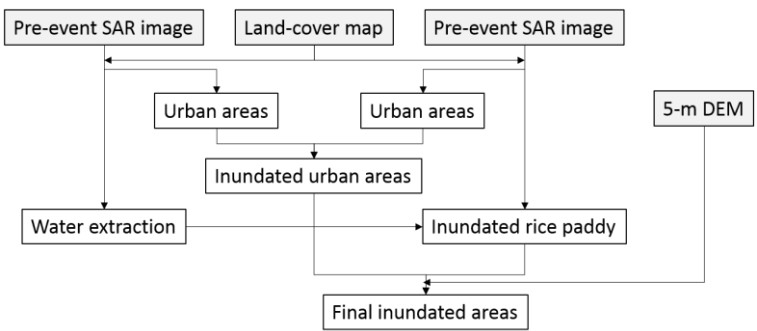

**Figure 7: Flowchart for the extraction of inundation areas**

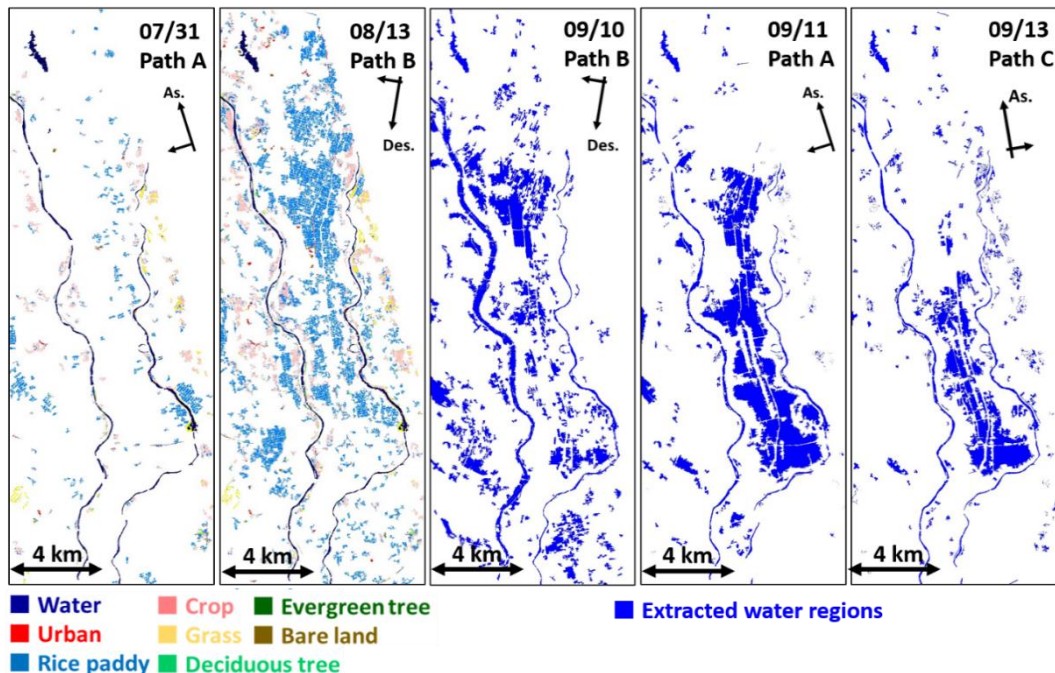

**Figure 8: Land-covers extracted using the proposed threshold values from the two pre-event PALSAR-2 images and the extracted water regions from the three co-event images after applying land-cover masks (corresponding to the low backscatter non-water regions).**

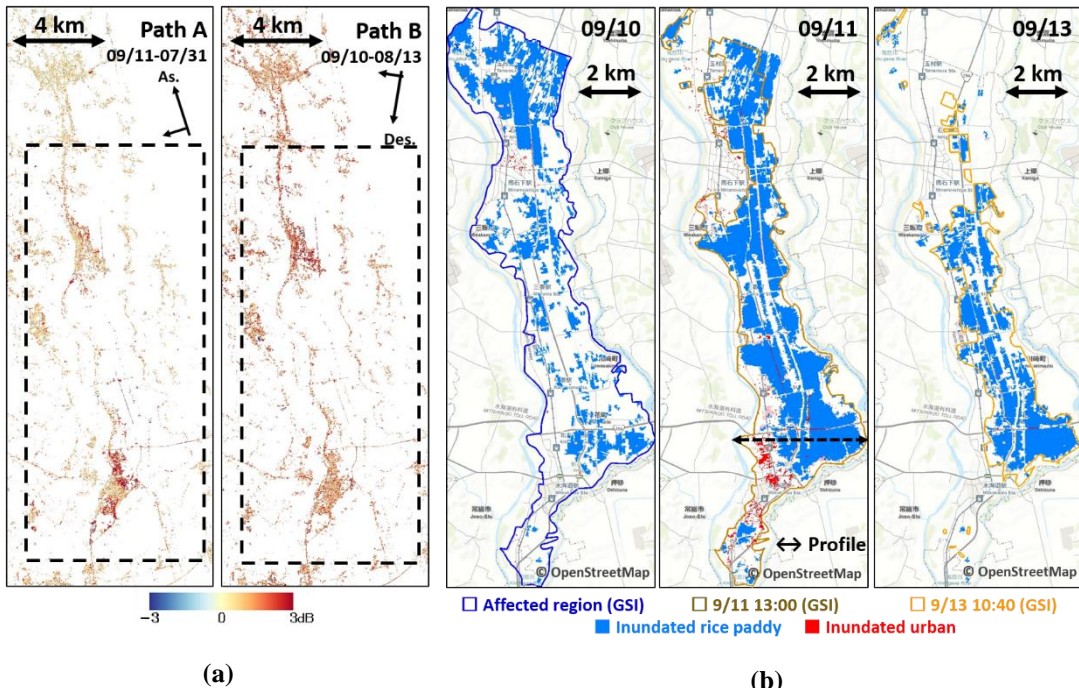

**Figure 9: (a) Differences in the backscattering coefficients for paths A and B within the urban area; (b) comparison of the extracted results and the visual interpretations overlapped on the GIS map, where blue pixels are flooded paddy fields, and the red pixels are flooded urban areas.**

**Table 3: Final extracted results for the inundation of the plain between the Kinugawa and the Kokai rivers and their accuracies in comparison with the reference data produced by GSI.**

| Date | GSI [km$^2$] | PALSAR [km$^2$] | Producer acc. | User acc. | Overall acc. | Kappa |
|------|-------------|----------------|---------------|-----------|--------------|-------|
| 9/11 | 30.5 | 21.7 | 65.6% | 92.2% | 69.6% | 0.36 |
| modified | | 30.2 | 85.1% | 86.1% | 78.3% | 0.41 |
| 9/13 | 15.2 | 10.7 | 61.0% | 87.0% | 81.8% | 0.59 |

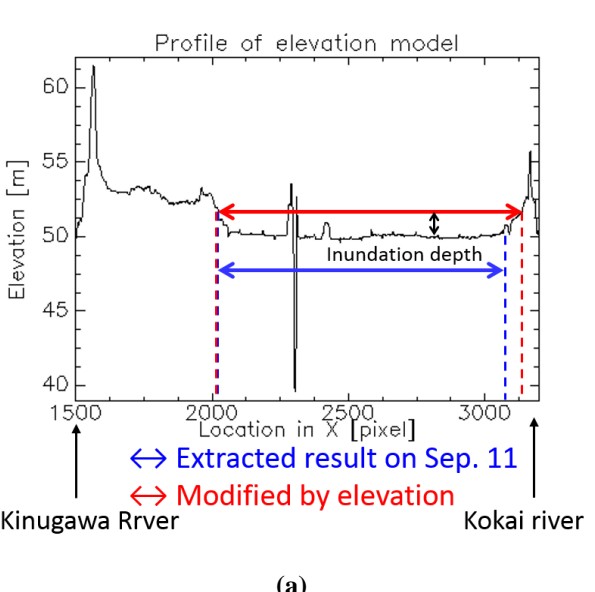

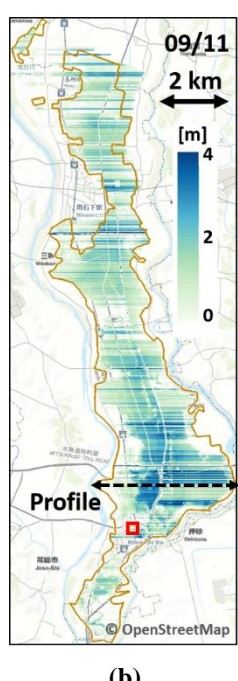

(a)                                                         (b)

**Figure 10: (a) Initial extracted inundation results (the blue arrow shows the range) for Sep. 11, 2015, and the modified result (the red arrow shows the range) based on the 5-m DEM profile. This modification was carried out on all the pixels from north to south for the rivers' transverse direction as an example of the west-east line shown in Figure 2(a). (b) The inundation depths on Sep. 11 as calculated from the modified results and DEM; the red square shows the location of City Hall.**

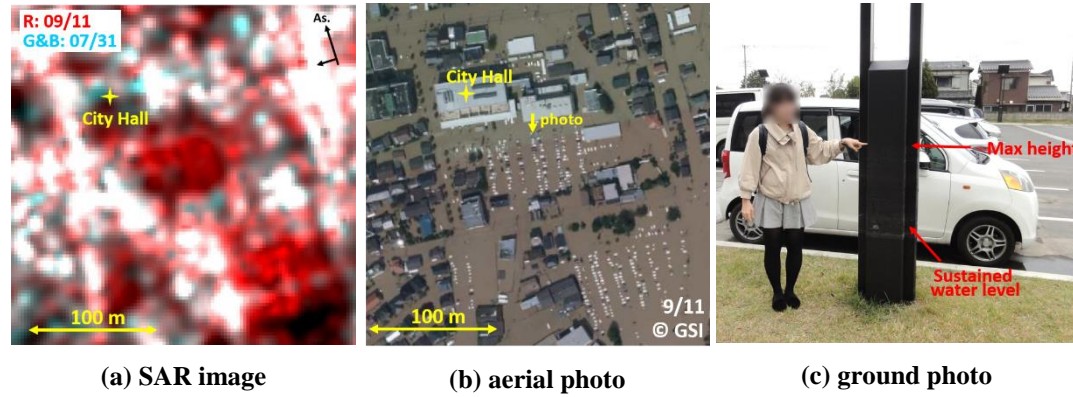

| (a) SAR image | (b) aerial photo | (c) ground photo |

**Figure 11: Close-up of the color composite of the SAR image for the red square in Figure 9(b) (a) and the co-event aerial photo (b) (GSI, 2015), which shows the surroundings of City Hall; a ground photo (c) taken in front of City Hall, where the max inundation height and the sustained water level can be confirmed by the water marks.**

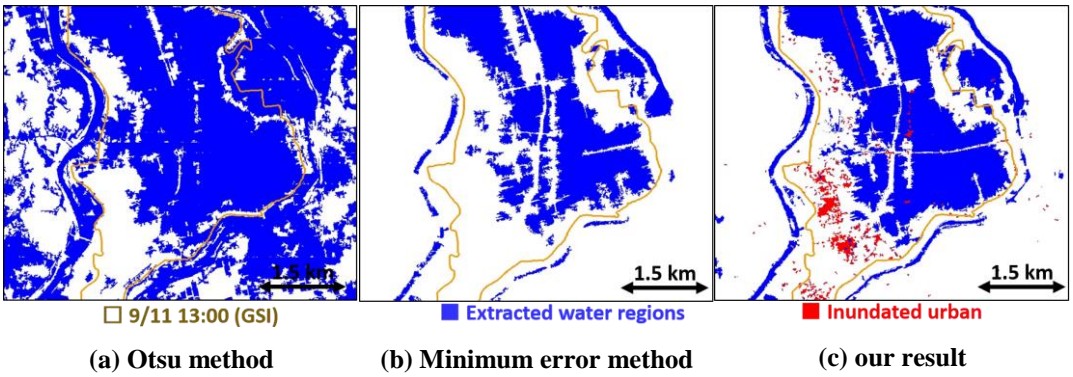

**(a) Otsu method**  **(b) Minimum error method**  **(c) our result**

Figure 12: Comparison of results by the automated thresholding methods: Otsu (a), Minimum error (b) and our thresholding result (c), around the profile line shown in Figure 10 (b).