# Peer review of "Detection of inundation areas due to the 2015 Kanto and Tohoku torrential rain in Japan based on multi-temporal ALOS-2 imagery"

_Natural Hazards and Earth System Sciences, 2018_

## Referee Comment (RC1) · Anonymous Referee #1 · 12 Apr 2018

The authors present a study that uses remote sensing techniques to detect inundated areas in Joso city, Japan after torrential rainfall in 2015. The manuscript is well written, interesting and scientifically sound. The study builds up on a previous study of the authors that uses the same satellite imagery, but seems to deploy a different detection method (Yamazaki and Liu 2016). The authors mention their previous study briefly in the manuscript, but it is not clear how this study differs from the previous one. Differences in method and results as well as the added-value of the new work would need to be clearly outlined in the manuscript.

Introduction: I strongly suggest adding a more in-depth review of the state-of-the-art on

inundation mapping from SAR images and how your study differs from other existing ones. More emphasize should also be given in presenting other studies that focussed on the same disaster and study area (if any). Based on this and a review of work that has been done related to inundation mapping from SAR images, it would be important to outline clearly the objectives of this study, the added-value that it can bring to improve existing inundation detection methods and the scientific understanding of the flood disaster.

Discussion: The manuscript would strongly benefit from a separate discussion section that clearly outlines the limitations and benefits of the applied method, and compares the results with findings of other studies (in particular your previous study).

Page 1, line 16: "...good level of agreement." Suggest replacing it with a more quantitative statement that mentions the actual accuracy metrics that you have computed.

Page 2, lines 11-19: Suggest moving this paragraph to Chapter 2 (Study area).

Page 4, line 16: "washed way" should be "washed away".

Page 5, line 2: "an SAR image" should be "a SAR image".

Page 5, lines 2-9: Suggest moving this paragraph to Chapter 1 (Introduction) as part of the state-of-the-art.

Page 7, line 21: Could not find "Figure 3(b)". Please check the figure references.

Page 8, line 5: I suggest adding here also a quantitative comparison with the results of your previous study. This would be needed to justify the mentioned improvements (Page 2, line 22).

---

## Referee Comment (RC2) · G. J.-P. Schumann (Referee) · 23 Apr 2018

This paper describes the application of ALOS PALSAR-2 images for flood mapping after a torrential rainfall in areas in Japan. The paper is generally well written and follows a clear structure, however I question somewhat the novelty and value of this study based on my comments below. I recommend carefully addressing all those comments.

- Reference other studies on flood mapping algorithm for SAR. I feel the authors missed many of those recently published.

- Better explain in how far this method is different from the procedure employed by the

authors in a previous study

- The authors show how the thresholds are selected but it seems to me this is all based on a rather manual technique requiring auxiliary data such as land use. I like the approach but I think, especially in the light of the many recent SAR-based techniques for flood mapping that are fully or semi-automated, the authors should clearly refer to those studies as well and also consider at least applying some of those for comparison. Furthermore, the authors need to justify why their rather simple, manual method should be preferred.

- For the urban area detection, the authors base this on intensity difference but I feel this should be done using coherence information. A couple of papers were recently published using coherence information to attempt mapping flooding in urban areas. In my opinion, using intensity and how this is done, is unclear to me and I question the validity of this part. The mapping results may be OK but then I doubt that the method presented here can be extrapolated to other areas.

---

## Author Comment (AC1) · 11 May 2018

**Detection of inundation areas due to the 2015 Kanto and Tohoku torrential rain in Japan based on multi-temporal ALOS-2 imagery**

Wen Liu[1], Fumio Yamazaki[1]

[1] Department of Urban Environment Systems, Chiba University, Chiba, 263-8522, Japan

*Correspondence to*: Wen Liu (wen.liu@chiba-u.jp)

*Our responses to reviewers' comments are written in Italic letters. Red letters are sentences reflected in the revised paper taking the comments.*

The authors present a study that uses remote sensing techniques to detect inundated areas in Joso city, Japan after torrential rainfall in 2015. The manuscript is well written, interesting and scientifically sound. The study builds up on a previous study of the authors that uses the same satellite imagery, but seems to deploy a different detection method (Yamazaki and Liu 2016). The authors mention their previous study briefly in the manuscript, but it is not clear how this study differs from the previous one. Differences in method and results as well as the added-value of the new work would need to be clearly outlined in the manuscript.

*Response: Thank you for your kind comments. We revised the manuscript as "One threshold value on backscattering intensity was investigated using the pre-event water regions and the pre-event PALSAR-2 images, and it was applied to all co-event images. In addition, the obtained results were only verified via visual comparison. In this study, the method of the inundation extraction was improved by introducing land-cover information and elevation data. The flooded urban areas were also extracted using the intensity difference between the pre- and co-event images. The obtained results are verified quantitatively via comparison with those from visual inspections of airborne imagery."*

**Introduction:** I strongly suggest adding a more in-depth review of the state-of-the-art on inundation mapping from SAR images and how your study differs from other existing ones. More emphasize should also be given in presenting other studies that focused on the same disaster and study area (if any). Based on this and a review of work that has been done related to inundation mapping from SAR images, it would be important to outline clearly the objectives of this study, the added-value that it can bring to improve existing inundation detection methods and the scientific understanding of the flood disaster.

*Response: We reorganized the introduction and added more descriptions of the previous study accordingly. Most of the previous studies on inundation mapping using X- or C-band SAR images. Due to the long wavelength, inundation extraction from L-band SAR images is one challenging point in this study.*

**Discussion:** The manuscript would strongly benefit from a separate discussion section that clearly outlines the limitations and benefits of the applied method, and compares the results with findings of other studies (in particular your previous study).

*Response: We will add a new chapter for discussion. The validity of introducing land-cover map will be verified. In addition, a comparison of the proposed method and an automated thresholding method will be added in the new chapter.*

Page 1, line 16: ": : :good level of agreement." Suggest replacing it with a more quantitative statement that mentions the actual accuracy metrics that you have computed.

*Response: According to the comment, we revised it as "more than 85% of the maximum inundation areas were extracted successfully".*

Page 2, lines 11-19: Suggest moving this paragraph to Chapter 2 (Study area).

*Response: According to the comment, we revised this part to the beginning of Chapter 2.*

Page 4, line 16: "washed way" should be "washed away".

*Response: It has been revised accordingly.*

Page 5, line 2: "an SAR image" should be "a SAR image".

*Response: It has been revised accordingly.*

Page 5, lines 2-9: Suggest moving this paragraph to Chapter 1 (Introduction) as part of the state-of-the-art.

*Response: According to the comment, we revised this part to Chapter 1.*

Page 7, line 21: Could not find "Figure 3(b)". Please check the figure references.

*Response: It has been revised to Figure 2(c).*

Page 8, line 5: I suggest adding here also a quantitative comparison with the results of your previous study. This would be needed to justify the mentioned improvements (Page 2, line 22).

*Response: According to the comment, we added a paragraph as "In the previous study (Yamazaki and Liu, 2016), the inundation areas in the three co-event PALSAR images were extracted using one threshold value of -12.4 dB, which was estimated by comparing the backscatter intensity for the original water regions (Kinugawa and Kogai rivers, Sanuma lake) and the other areas in the whole study area. As a result, 20.4 km² on September 11 and 16.3 km² on September 13 were extracted as inundations. Since the threshold values used in this study were -13.5 dB for September 11 and -14.2 dB respectively, lower than the previous study, the extracted areas including the inundated built-up areas were similar in size to that of the previous results. However, the producer and user accuracies increased 3%, whereas the O.A. increased 2% for the results on September 11. For the results on September 13, the producer accuracy decreased whereas the user accuracy increased from 68.8% to 87%. The O.A increased significantly from 77.4% to 81.3%, while the kappa coefficient increased from 0.53 to 0.58. The induvial threshold values for the images taken in different acquisition conditions were more effective than one common value."*

---

## Author Response (AR1)

**Detection of inundation areas due to the 2015 Kanto and Tohoku torrential rain in Japan based on multi-temporal ALOS-2 imagery**

Wen Liu[1], Fumio Yamazaki[1]

[1] Department of Urban Environment Systems, Chiba University, Chiba, 263-8522, Japan

*Correspondence to*: Wen Liu (wen.liu@chiba-u.jp)

*Our responses to reviewers' comments are written in Italic letters. Red letters are sentences reflected in the revised paper taking the comments.*

Reviewer 1

The authors present a study that uses remote sensing techniques to detect inundated areas in Joso city, Japan after torrential rainfall in 2015. The manuscript is well written, interesting and scientifically sound. The study builds up on a previous study of the authors that uses the same satellite imagery, but seems to deploy a different detection method (Yamazaki and Liu 2016). The authors mention their previous study briefly in the manuscript, but it is not clear how this study differs from the previous one. Differences in method and results as well as the added-value of the new work would need to be clearly outlined in the manuscript.

*Response: Thank you for your kind comments. We revised the manuscript as "In the study, one threshold value of backscattering intensity was investigated using the pre-event water regions and the pre-event PALSAR-2 images, and it was applied to all co-event images. In addition, the obtained results were only verified via visual comparison. In this study, the method of the inundation extraction is improved by introducing land-cover information and elevation data. The flooded urban areas are also extracted using the intensity difference between the pre- and co-event images. The obtained results are verified quantitatively via comparison with those from visual inspections of airborne imagery." in **1. Introduction**.*

*The comparison with the previous study has been added in **5.2 Verification and improvement** as "In the previous study (Yamazaki and Liu, 2016), the inundation areas in the three co-event PALSAR images were extracted using one threshold value of -12.4 dB, which was estimated by comparing the backscatter intensity for the original water regions (Kinugawa and Kogai rivers, Sanuma lake) and the other areas in the whole study area. As a result, 20.4 km2 on September 11 and 16.3 km2 on September 13 were extracted as inundation. Since the threshold values used in this study were -13.5 dB for September 11 and -14.2 dB for September 12, respectively, lower than that of the previous study, the extracted areas including the inundated built-up areas were similar in size to that of the previous results. However, the producer and user accuracies increased 3%, whereas the O.A. increased 2% for the result on September 11. For the result on September 13, the producer accuracy decreased whereas the user accuracy increased from 68.8% to 87%. The O.A. increased significantly from 77.4% to 81.3%, while the kappa coefficient increased from 0.53 to 0.58. The individual threshold values for the images taken in different acquisition conditions were more effective than one common value."*

**Introduction:** I strongly suggest adding a more in-depth review of the state-of-the-art on inundation mapping from SAR images and how your study differs from other existing ones. More emphasize should also be given in presenting other studies that focused on the same disaster and study area (if any). Based on this and a review of work that has been done related to inundation mapping from SAR images, it would be important to outline clearly the objectives of this study, the added-value that it can bring

to improve existing inundation detection methods and the scientific understanding of the flood disaster.

*Response: We revised the description of the state-of-the-art part in the introduction as "Because microwaves exhibit specular reflections against a smooth water surface, water regions in a SAR image show low backscattering intensity. SAR images are effective for extracting inundation areas. Several methods, both pixel- and object-based, have been proposed to extract inundation zones from SAR images (Martinis et al., 2009 and 2013; Hoque et al., 2011; Manjusree et al., 2012; Pulvirenti et al., 2014; Kundu et al., 2015; Nakmuenwai et al., 2017). Thresholding is a common and effective pixel-based approach. Since backscattering of a water surface depends on many factors such as acquisition conditions of SAR images and their environments, its value is highly variable. It is difficult to judge the most suitable value objectively without additional information. Automated thresholding methods using the gray-level histogram have been introduced to overcome this issue (Fan and Lei, 2009; Martinis 2009 and 2013; Pulvirenti et al., 2011; Nakmuenwai et al., 2017). The global threshold value was merged from several local threshold values, which were obtained from the multimodal histograms of sub-areas. However, this approach is time-consuming when the study area is large. In addition, sufficient contrast was necessary for automated thresholding. Giustarini et al. (2013) found the previous proposed methods were difficult to work in urban areas containing radar shadow and layover. They proposed a method based on image differencing to detect floodwater inside urban areas. Mason et al. (2009 and 2012) used a SAR simulator and Lidar data to estimated inundated buildings. Interfeormatic coherence was also used to extract floods in either rural or urban areas, but the acquisitions of temporal and spatial baselines were strict (Nico et al. 2000; Chini et al. 2012; Pulvirenti et al. 2016). All of these researches used SAR images taken by X- and C- bands with short wavelengths, which were sensitive to separate water and non-water regions. Flood mapping using L-band satellite images was few (Zhang and Wang 2003; Allan et al., 2012; Yulianto et al. 2015)".*

As mentioned in the new manuscript, most of the previous studies on inundation mapping using X- or C-band SAR images. Due to the long wavelength, inundation extraction from L-band SAR images is one challenging point in this study.

The comparison of the previous study for the same event was added in the new chapter **6. Discussions** *"To verify the effectiveness of our results, a comparison with the previous studies for the same event was carried out. Natsuaki et al. (2016) proposed a combination of coherence and amplitude values to detect affected areas using two pre-event and one co-event PALSAR-2 images taken on September 12, 2015. Inundation was extracted by the decrease of coherence and a low backscatter intensity. Kwak et al. (2017) extracted the floods on September 11 from a pair of pre-event and co-event PALSAR-2 images taken on July 31 and September 11, 2015, which were also used in our study. Flooded rice paddies were extracted by the differences of intensity whereas flooded urban areas were extracted by the correlation coefficient. These two researches extracted both the inundated rice paddies and urban areas using only SAR images. The producer accuracy in the study of Natsuaki et al. (2016) was 75%, a little higher than our results before the improvement using DEM. However, the O. A. was 52% since some areas could not be evaluated due to low pre-event coherence values. Our method using only the backscatter intensity could be applied to the whole study area. The accuracy in the study of Kwak et al. (2017) was not indicated. By visual comparison, their results extracted more inundated areas with more commission errors. Many agriculture fields outside the inundation were extracted as false alarms. Rimba and Miura (2017) compared three common methods, unsupervised and supervised classifications, threshold method using the same SAR pair of Kwak et al. (2017) and 5-m DEM. The scheme of threshold method showed the best result, which extracted water regions from the pre- and co-event images, respectively, similar with our proposed approach. The inundation was obtained by the change of the extracted water regions. However, their scheme would not work when the inundated rice paddies were poured in a pre-event image. Our method could overcome this problem by applying the land cover map. The inundated urban areas were not extracted in the Rimba and Miura's research."*

**Discussion:** The manuscript would strongly benefit from a separate discussion section that clearly outlines the limitations and benefits of the applied method, and compares the results with findings of other studies (in particular your previous study).

*Response: We added a new chapter **6. Discussion** accordingly.*

*The limitations and benefits of the proposed method is descripted in the new chapter as "Although both the pre- and co-event SAR images were used in this study, the extraction of water region was carried out for each image. When a pre-event image is not available, an inundation map can still be created by the thresholding method. Inundated urban areas can be extracted by the change detection, however, a pre-event image taken in the same path is necessary. A land-cover map was introduced to define the threshold values and to reduce commission errors in a smooth surface due to a longer L-band wavelength. In the thresholding approach, the land cover could be replaced by visual interpretation. Without a land cover map, commission errors in the inundation extraction would decrease the accuracy. The 5-m DEM was used for improving the inundation map and for estimating the inundation depth. In an emergency response phase, our method could still obtain a reasonable result with the overall accuracy higher than 70%. Thus, our method is still valid if a land-cover map and detailed DEM data are not available. However, the accuracy of the obtained inundation map would decrease."*

*The comparison of the proposed method and an automated thresholding method was also added in the new chapter as "Several common automated thresholding algorithms were applied to the PALSAR-2 image on September 11 to compare with our results (Kapur et al., 1985; Ridler and Calvard, 1978; Kittler and Illingworth, 1986; Otsu, 1979). Most of the automated algorithms extracted the inundation excessively. **Figure 12** shows the comparison of two best results by the Otsu (1979) and Minimum error thresholding algorithms (Kittler and Illingworth, 1986) and our thresholding result. In this enlarged region, the overall accuracy for the three results were calculated. Our results using the threshold value -13.5 dB obtained the highest accuracy as 74%, whereas that for the Otsu method was 55% and for the Minimum error method was 72%. After merging the extracted urban areas, the overall accuracy increased to 76%."*

Page 1, line 16: ": : :good level of agreement." Suggest replacing it with a more quantitative statement that mentions the actual accuracy metrics that you have computed.

*Response: According to the comment, we revised it as "more than 85% of the maximum inundation areas were extracted successfully".*

Page 2, lines 11-19: Suggest moving this paragraph to Chapter 2 (Study area).

*Response: According to the comment, we revised this part to the beginning of Chapter 2.*

Page 4, line 16: "washed way" should be "washed away".

*Response: It has been revised accordingly.*

Page 5, line 2: "an SAR image" should be "a SAR image".

*Response: It has been revised accordingly.*

Page 5, lines 2-9: Suggest moving this paragraph to Chapter 1 (Introduction) as part of the state-of-the-art.

*Response: According to the comment, we revised this part to **1. Introduction**.*

Page 7, line 21: Could not find "Figure 3(b)". Please check the figure references.

*Response: It has been revised as "Figure 2(c)".*

Page 8, line 5: I suggest adding here also a quantitative comparison with the results of your previous study. This would be needed to justify the mentioned improvements (Page 2, line 22).

*Response: As the answer of the general comment, the comparison with the previous study has been added in **5.2 Verification and improvement**.*

**Reviewer 2**

This paper describes the application of ALOS PALSAR-2 images for flood mapping after a torrential rainfall in areas in Japan. The paper is generally well written and follows a clear structure, however I question somewhat the novelty and value of this study based on my comments below. I recommend carefully addressing all those comments.

*Response: Thank you for your kind comments.*

- Reference other studies on flood mapping algorithm for SAR. I feel the authors missed many of those recently published.

*Response: The introduction was reorganized and more references were added as "Because microwaves exhibit specular reflections against a smooth water surface, water regions in a SAR image show low backscattering intensity. SAR images are effective for extracting inundation areas. Several methods, both pixel- and object-based, have been proposed to extract inundation zones from SAR images (Martinis et al., 2009 and 2013; Hoque et al., 2011; Manjusree et al., 2012; Pulvirenti et al., 2014; Kundu et al., 2015; Nakmuenwai et al., 2017). Thresholding is a common and effective pixel-based approach. Since backscattering of a water surface depends on many factors such as acquisition conditions of SAR images and their environments, its value is highly variable. It is difficult to judge the most suitable value objectively without additional information. Automated thresholding methods using the gray-level histogram have been introduced to overcome this issue (Fan and Lei, 2009; Martinis 2009 and 2013; Pulvirenti et al., 2011; Nakmuenwai et al., 2017). The global threshold value was merged from several local threshold values, which were obtained from the multimodal histograms of sub-areas. However, this approach is time-consuming when the study area is large. In addition, sufficient contrast was necessary for automated thresholding. Giustarini et al. (2013) found the previous proposed methods were difficult to work in urban areas containing radar shadow and layover. They proposed a method based on image differencing to detect floodwater inside urban areas. Mason et al. (2009 and 2012) used a SAR simulator and Lidar data to estimated inundated buildings. Interfeormatic coherence was also used to extract floods in either rural or urban areas, but the acquisitions of temporal and spatial baselines were strict (Nico et al. 2000; Chini et al. 2012; Pulvirenti et al. 2016). All of these researches used SAR images taken by X- and C- bands with short wavelengths, which were sensitive to separate water and non-water regions. Flood mapping using L-band satellite images was few (Zhang and Wang 2003; Allan et al., 2012; Yulianto et al. 2015)".*

*There previous studies for the same event were discussed in the new chapter **6. Discussions** as "To verify the effectiveness of our results, a comparison with the previous studies for the same event was carried out. Natsuaki et al. (2016) proposed a combination of coherence and amplitude values to detect affected areas using two pre-event and one co-event PALSAR-2 images taken on September 12, 2015. Inundation was extracted by the decrease of coherence and a low backscatter intensity. Kwak et al. (2017) extracted the floods on September 11 from a pair of pre-event and co-event PALSAR-2 images taken on July 31 and September 11, 2015, which were also used in our study. Flooded rice paddies were extracted by the differences of intensity whereas flooded urban areas were extracted by the correlation coefficient. These two researches extracted both the inundated rice paddies and urban areas using only SAR images. The producer accuracy in the study of Natsuaki et al. (2016) was 75%, a little higher than our results before the improvement using DEM. However, the O. A. was 52% since some areas could not be evaluated due to low pre-event coherence values. Our method using only the backscatter intensity could be applied to the whole study area. The accuracy in the study of Kwak et al. (2017) was not indicated. By visual comparison, their results extracted more inundated areas with more commission errors. Many agriculture fields outside the inundation were extracted as false alarms.*
*Rimba and Miura (2017) compared three common methods, unsupervised and supervised classifications, threshold method using*

*the same SAR pair of Kwak et al. (2017) and 5-m DEM. The scheme of threshold method showed the best result, which extracted water regions from the pre- and co-event images, respectively, similar with our proposed approach. The inundation was obtained by the change of the extracted water regions. However, their scheme would not work when the inundated rice paddies were poured in a pre-event image. Our method could overcome this problem by applying the land cover map. The inundated urban areas were not extracted in the Rimba and Miura's research."*

- Better explain in how far this method is different from the procedure employed by the authors in a previous study

*Response: According to the comment, the comparison with our previous study has been added in **5.2 Verification and improvement** as "In the previous study (Yamazaki and Liu, 2016), the inundation areas in the three co-event PALSAR images were extracted using one threshold value of -12.4 dB, which was estimated by comparing the backscatter intensity for the original water regions (Kinugawa and Kogai rivers, Sanuma lake) and the other areas in the whole study area. As a result, 20.4 km2 on September 11 and 16.3 km2 on September 13 were extracted as inundations. Since the threshold values used in this study were -13.5 dB for September 11 and -14.2 dB respectively, lower than the previous study, the extracted areas including the inundated built-up areas were similar in size to that of the previous results. However, the producer and user accuracies increased 3%, whereas the O.A. increased 2% for the results on September 11. For the results on September 13, the producer accuracy decreased whereas the user accuracy increased from 68.8% to 87%. The O.A increased significantly from 77.4% to 81.3%, while the kappa coefficient increased from 0.53 to 0.58. The induvial threshold values for the images taken in different acquisition conditions were more effective than one common value."*

- The authors show how the thresholds are selected but it seems to me this is all based on a rather manual technique requiring auxiliary data such as land use. I like the approach but I think, especially in the light of the many recent SAR-based techniques for flood mapping that are fully or semi-automated, the authors should clearly refer to those studies as well and also consider at least applying some of those for comparison. Furthermore, the authors need to justify why their rather simple, manual method should be preferred.

*Response: The comparison of the proposed method and an automated thresholding method was added in the new chapter **6. Discussions** as "Several common automated thresholding algorithms were applied to the PALSAR-2 image on September 11 to compare with our results (Kapur et al., 1985; Ridler and Calvard, 1978; Kittler and Illingworth, 1986; Otsu, 1979). Most of the automated algorithms extracted the inundation excessively. **Figure 12** shows the comparison of two best results by the Otsu (1979) and Minimum error thresholding algorithms (Kittler and Illingworth, 1986) and our thresholding result. In this enlarged region, the overall accuracy for the three results were calculated. Our results using the threshold value -13.5 dB obtained the highest accuracy as 74%, whereas that for the Otsu method was 55% and for the Minimum error method was 72%. After merging the extracted urban areas, the overall accuracy increased to 76%."*

- For the urban area detection, the authors base this on intensity difference but I feel this should be done using coherence information. A couple of papers were recently published using coherence information to attempt mapping flooding in urban areas. In my opinion, using intensity and how this is done, is unclear to me and I question the validity of this part. The mapping results may be OK but then I doubt that the method presented here can be extrapolated to other areas.

*Response: Since coherence is calculated from two temporal images, more than 3 temporal images (two pre- and one post-event) taken in the same acquisition condition are necessary for change detection. Thus, we could not use coherence in this study.*

*We added the references for the detection of inundated urban areas in **1. Introduction** as "Mason et al. (2009 and 2012) used a SAR simulator and Lidar data to estimated inundated buildings. Interfeormatic coherence was also used to extract floods in either rural or urban areas, but the acquisitions of temporal and spatial baselines were strict (Nico et al. 2000; Chini et al. 2012; Pulvirenti et al. 2016)." and in **6. Discussions** as "Natsuaki et al. (2016) proposed a combination of coherence and amplitude values to detect affected areas using two pre-event and one co-event PALSAR-2 images taken on September 12, 2015. Inundation was extracted by the decrease of coherence and a low backscatter intensity. Kwak et al. (2017) extracted the floods on September 11 from a pair of pre-event and co-event PALSAR-2 images taken on July 31 and September 11, 2015, which were also used in our study. Flooded rice paddies were extracted by the differences of intensity whereas flooded urban areas were extracted by the correlation coefficient. These two researches extracted both the inundated rice paddies and urban areas using only SAR images."*

*The limitations and benefits of the proposed method was discussed in the new chapter **6. Discussions** as "Although both the pre- and co-event SAR images were used in this study, the extraction of water region was carried out for each image. When a pre-event image is not available, an inundation map can still be created by the thresholding method. Inundated urban areas can be extracted by the change detection, however, a pre-event image taken in the same path is necessary. A land-cover map was introduced to define the threshold values and to reduce commission errors in a smooth surface due to a longer L-band wavelength. In the thresholding approach, the land cover could be replaced by visual interpretation. Without a land cover map, commission errors in the inundation extraction would decrease the accuracy. The 5-m DEM was used for improving the inundation map and for estimating the inundation depth. In an emergency response phase, our method could still obtain a reasonable result with the overall accuracy higher than 70%. Thus, our method is still valid if a land-cover map and detailed DEM data are not available. However, the accuracy of the obtained inundation map would decrease."*